

# Identification and comparison of circular RNAs in preeclampsia

Zepeng Ping[1,*], Ling Ai[1,*], Huaxiang Shen[1], Xing Zhang[2], Huling Jiang[1] and Ye Song[3]

[1] Department of Obstetrics, Maternity and Child Health Care Affiliated Hospital, Jiaxing University, Jiaxing, China
[2] School of Biology & Basic Medical Science, Soochow University, Suzhou, China
[3] Department of Obstetrics, Suzhou Municipal Hospital, Suzhou, Jiangsu, China
[*] These authors contributed equally to this work.

## ABSTRACT

**Background.** Preeclampsia (PE) is a pregnancy-specific syndrome, belongs to the gestational hypertension diseases category and is considered among the causes of maternal and perinatal mortality and morbidity. However, the pathogenesis of PE is still vague.

**Methods.** In the present study, the circular RNA (circRNA) expression patterns of normal pregnant women and PE patients were investigated using whole RNA sequencing.

**Results.** A total of 151 differential expressed circRNAs were identified including 121 upregulated and 30 downregulated ones. Functional and pathway enrichment analysis was conducted on the differentially expressed circRNAs using Gene Ontology and KEGG databases. The results of this analysis indicated that several crucial biological processes and pathways were enriched in PE patients. circRNA–microRNA (miRNA) interaction analysis indicated that the reported differentially expresse circRNAs may be associated with some regulatory functions through miRNAs in PE patients. Two ceRNAs networks were constructed according to the targeting relationship between circRNAs/miRNAs and miRNAs/mRNAs. One sub-network contained one upregulated circRNA, four downregulated miRNAs and five upregulated mRNAs, and another sub-network contained 10 downregulated circRNAs, 21 upregulated miRNAs and 15 downregulated mRNAs.

**Conclusion.** CircRNA expression patterns have been investigated and this analysis revealed their potential regulatory mechanisms in PE patients. We constructed the ceRNAs (competing endogenous RNA) to reveal the potential molecular roles of dysregulated circRNAs in the PE patients using RNA sequencing data. circRNA_13301 was the only one upregulated circRNA in ceRNA being targeted by four miRNAs.

Corresponding authors
Huling Jiang, jianghuling@zju.edu.cn
Ye Song,
202061224050@njtech.edu.cn

## INTRODUCTION

Preeclampsia (PE) is a pregnancy-specific syndrome, belongs to the common gestational hypertension disease category, and is considered among the causes of mortality and

morbidity of pregnant women (*Cote et al., 2008*; *Jia & Li, 2019*). A pregnant women is diagnosed with PE when the both of following clinical features are identified of systolic/diastolic blood pressure ≥140/≥90 mmHg and substantial proteinuria ≥300 mg during 24 h at or after 20 weeks of gestation (*Milne et al., 2005*; *Wisner, 2019*). Women with severe PE might present multisystem disturbance syndrome (*Maynard et al., 2005*; *Abalos et al., 2013*). The exact etiologyand pathogenesis of PE still remains unclear.

Numerous recent studies have illustrated that a variety of non-coding RNAs (ncRNAs) are related to the pathogenesis of pregnancy diseases (*Huang et al., 2018*; *Jiao, Wang & Huang, 2018*; *Wang et al., 2018*; *Cheng et al., 2019*; *Yang et al., 2019*), wherein, circular RNAs (circRNAs) belong to one type of ncRNAs with various of biological features, such as evolutionary conservation, structural stability and tissue specificity so on (*Jeck et al., 2013*; *Memczak et al., 2013*), which play crucial roles in biological functions of certain diseases (*Jia & Li, 2019*). The biological roles of circRNAs were gradually exposed from studies on various cancers and many other diseases, and the prevalent hypothesis for their roles include the following: miRNA sponges, RNA-binding proteins (RBPs), gene transcription regulation and mRNA translation mediators (*Kristensen et al., 2019*). Some dysregulated circRNAs were associated with the development and the progression of numerous diseases (*Bai et al., 2019*; *Li et al., 2019a*; *Li et al., 2019b*; *Zhou et al., 2019*), while other dysregulated circRNAs have presented the potential to be used as biomarkers (*Shao et al., 2019*; *Tian et al., 2019*; *Wang et al., 2019*; *Wu et al., 2019*). In regard to the role of circRNAs in PE, several studies have been reported from other groups, who identified crucial roles of specific circRNAs in the pathogenesis of PE by regulating ceRNA. Furthermore, some circRNAs have been suggested as potential novel biomarkers for PE. Most of these circRNAs were identified from the trophoblast cell or human placenta using human circRNA microarray or using publicly available datasets from the Gene Expression Omnibus (GEO) databaseand meta-analyzing them with bioinformatics techniques (*Zhou et al., 2018*; *Liu et al., 2019*). In addition, Hsa_circ_0007121 was reported to mediate the progression of PE via miR-182-5p/placental growth factor (PGF) axis (*Gai et al., 2020*). CircLRRK1 was identified to suppress the proliferation, migration and invasion of trophoblast cells via miR-223-3p/PI3K/AKT axis (*Tang, Zhang & Han, 2020*). Circ_0001438 aggravated the dysfunctions of human villous trophoblasts by mediating the miR-942/NLRP3 axis (*Li et al., 2020*). circCRAMP1L (*Zhang et al., 2020b*; *Zhang et al., 2020a*), CircSFXN1 (*Zhang et al., 2020b*; *Zhang et al., 2020a*), Circ_0085296 (*Zhu et al., 2020*) were reported to play important roles in the pathogenesis of PE.

In the present study, the expression profiles of circRNAs in PE and healthy pregnant women were investigated with whole transcriptome sequencing and the regulatory mechanism of circRNAs in PE was explored. The findings of this study will contribute to the understanding of the etiology and the pathogenesis of PE, especially the regulatory mechanism of circRNAs in PE.

**Table 1  The information of patients with PE and normal control groups.**

| Parameter | PE ($n = 20$) | Healthy controls ($n = 20$) | P value |
|---|---|---|---|
| Age (years) | 27.59 ± 4.20 | 27.65 ± 2.45 | ns |
| Gestational weeks | 36.23 ± 2.81 | 36.88 ± 2.71 | ns |
| Systolic blood pressure (mmHg) | 168.91 ± 13.25 | 116.55 ± 13.18 | *** |
| Diastolic blood pressure (mmHg) | 109.91 ± 10.94 | 69.8 ± 9.89 | *** |
| Urine protein (g/24 h) | 2.73 ± 1.69 | 0.09 ± 0.06 | *** |

Notes.
The two groups were matched by age and gestational week. ns, $P > 0.05$ and ***, $P < 0.01$.

# MATERIALS & METHODS

## Ethics statement

This study was approved by the Ethics Committee of the Faculty of Medicine from the Maternity and Child Health Care Affiliated Hospital, Jiaxing University (Ethical Application Ref: 2019-47). Informed consent was signed in agreement with subjects in the study.

## Sample collection

A total of 40 pregnant women (20 cases with PE patients and 20 normal pregnant women) from the Maternity and Child Health Care Affiliated Hospital, Jiaxing University were recruited for this study (Table 1). All pregnant women were strictly screened according to the clinical features of PE cases to exclude other causes (*Wisner, 2019*). The whole blood samples (3.0 ml) were collected from the PE patients and normal pregnant women, and leukocyte cells were isolated from the collected whole blood using the Red Blood Cell Lysis Buffer kit (Tiangen, Beijing, China). The cells were transferred into Eppendorf tubes with TRIzol reagent (Invitrogen Carlsbad, CA, USA). Total RNA was quantified with a NanoDrop ND 2000 spectrophotometer and RNA integrity was assessed using an Agilent Bioanalyzer 2100 (Agilent Technologies Santa Clara, CA, USA).

## rRNA-depleted RNA sequencing

First, Total RNAs was extracted from three PE cases and three normal controls using the TRIzol reagent. After quality control, the TruSeq Stranded Total RNA with Ribo-Zero Gold kit (for plants, use the TruSeq Stranded Total RNA LT-(with Ribo-Zero Plant) kit ) was used to digest ribosomal RNA and added a breaking reagent to break the RNA into short fragments. Later the disrupted RNA was used as a template to synthesize one-strand cDNA with six-base random primers and then prepare a two-strand synthesis reaction system to synthesize two-strand cDNAwith dUTP instead of dTTP. Followed with the UNG enzymatic method to digest a strand containing dUTP, and only the first cDNA strand with different linkers in the linking strand was retained. The first cDNA strand was purified and repaired. A-tailed was connected to the sequencing adapter, and then fragment size was selected for finally PCR amplification. RNA Integrity Number threshold of 7 was applied to construct the libraries using TruSeq Stranded Total RNA Library Prep Kit (Illumina, USA). Six libraries were sequenced on the HiSeqTM4000 sequencing platform. Sequencing analysis was conducted by Shanghai OE Biotech, Shanghai, China. All raw data has been uploaded to Sequence Read Archieve: PRJNA665923.

### CircRNA prediction, expression analysis and interactions research

After performing quality control tests on the raw fastq files, the extracted clean reads were aligned to the human reference genome (GRCh38.p12) using hisat2 tool. The CIRI software (2015) was used to scan for paired chiastic clipping signals (CIRI: an efficient and unbiased algorithm for de novo circular RNA identification). circRNA sequences were predicted based on the junction reads and GT-AG cleavage signals. RPM algorithm was applied to calculate the relative expression of circRNAs normalizing the matched read number with the transcripts' length. Miranda software was used to predict the targeting interactions between miRNAs and circRNAs.

### Differential screening analysis and Functional Analysis

DESeq tool (http://www.bioconductor.org/packages/release/bioc/html/DESeq.html) was applied to perform differential expression analysis and to calculate the *p*-values and fold-change of each identified transcript. The differentially expressed (DE) circRNAs were screened with standard of *p* value threshold < 0.05 and fold change threshold > 2. Enrichment analysis of the DE circRNAs against the Gene Ontology (GO) and Kyoto Encyclopedia of Genes and Genomes (KEGG) was conducted with using the Hypergeometric Distribution Test method. DIANA-mirPath was utilized to predict miRNA targets.

### ceRNA network analysis

Based on the ceRNA (competing endogenous RNA) regulatory mechanisms (*Salmena et al., 2011*), the miRNA targets (miRNA-mRNA or miRNA-circRNA targets) were predicted, and then circRNA-mRNA regulation pairs were predicted according with the guilty by association method (*Zhang et al., 2017*; *Zhang, Zhu & Hu, 2018*). At the same time, the correlation of regulation relation to expression value was calculated. The ceRNA networks of upregulated circRNA–downregulated miRNA–upregulated mRNA and downregulated circRNA–upregualted miRNA–downregulated mRNA were constructed with Cytoscape software version 3.7.1.

### Real-time PCR assay

Seven DE circRNAs were randomly selected for validation with real-time PCR. The divergent primers were designed according to flanking sequences of junction sites of the circRNAs (Table 2). RNA was extracted for 20 PE samples and 20 normal control samples. cDNAs was produced as the template by cDNA Synthesis SuperMix (Transgen Biotech, Beijing, China). Real-time PCR was performed on Real-Time PCR Detection System (Bio-Rad,USA) with Universal SYBR Green Supermix (Bio-Rad, USA) according to the procedure. $\beta$-actin was used as internal standard control and delta delta CT method ($2^{-\Delta\Delta Ct}$) method was applied to calculate the expression level of circRNAs. All experiments were repeated in three times.

**Table 2   Primers for validation of circRNAs.**

| Primer name | Sequence (5′–3′) | Length (bp) |
| --- | --- | --- |
| hsa_circ_0001289-F | AACAGAGTCAGCATCAGAGC | 20 |
| hsa_circ_0001289-R | AGTGGCATCTATTATTGAA | 19 |
| hsa_circ_0008311-F | GTTTCTGGTTCTCAGGAT | 18 |
| hsa_circ_0008311-R | TCAATGAGAGGTCCCATCTGG | 21 |
| hsa_circ_0004960-F | TGAAAGCAAGTCACTAGAGAT | 20 |
| hsa_circ_0004960-R | CCAAAGAGGGAGCCAGATGT | 20 |
| hsa_circ_0003753-F | CTTGAAGTTTTACTACTGAG | 20 |
| hsa_circ_0003753-R | CCACGGTGACATTGGCTG | 18 |
| hsa_circ_0001861-F | AATCAAGAAGCGTGGGATCC | 20 |
| hsa_circ_0001861-R | TGCTCCACCTCACAGTTC | 18 |
| hsa_circ_0006719-F | TTACACCCAGTGCCTCTGAC | 20 |
| hsa_circ_0006719-R | TGTCCGCAGGCAACCCTT | 18 |
| hsa_circ_0005806-F | CGGCCACCTTTGAGGCTCT | 19 |
| hsa_circ_0005806-R | GAGAACAAGGAGGGGTGGTG | 20 |

# RESULTS

## Characterization of circRNAs from the PE and normal pregnant women

After the circRNA prediction with the CIRI software, the identified circRNAs were compared with human circRNAs reported in CircBase (http://www.circbase.org/) with 14674 of them being identified in both PE and healthy pregnant women. 6842 circRNAs had been recorded in CircBase data with 7832 circRNAs being newly introduced ones (Fig. 1A). The scaled relative expressions of circRNAs in the six samples of the present study were displayed with a heatmap (Fig. 1B). The $p$-values and fold changes reported from the differential expression analysis of circRNAs were visualized in the form of a volcano plot (Fig. 1C). After a stringent screening strategy, adjusting $p$-values for multiple testing (adjusted $p$ value < 0.05; fold change, >2 or < 0.5), 120 significantly upregulated and 31 significantly downregulated circRNAs were identified in the leukocytes of PE women compared with control groups (Fig. 1D). These findings indicated that the relative expression of several circRNAs was altered in the leukocytes of the pregnants with PE, indicating that these DE circRNAs may be associated with the etiology and pathogenesis of PE.

## GO annotation of DE circRNAs

To gain further insights into the biological processes that were potentially mediated by these DE circRNAs in PE, GO annotation analysis was used to determine the functions of differentially expressed circRNAs using the parental transcripts. 64 GO terms were significantly enriched (FDR < 0.05) in the source transcripts of the DE circRNAs. The top 30 biological processes related GO terms that were found to be enriched in upregulated circRNAs included terms such as negative regulation of cytokine-mediated signaling pathway, positive regulation of macrophage tolerance induction, negative regulation of macrophage cytokine production, response to peptidoglycan, regulation of protein

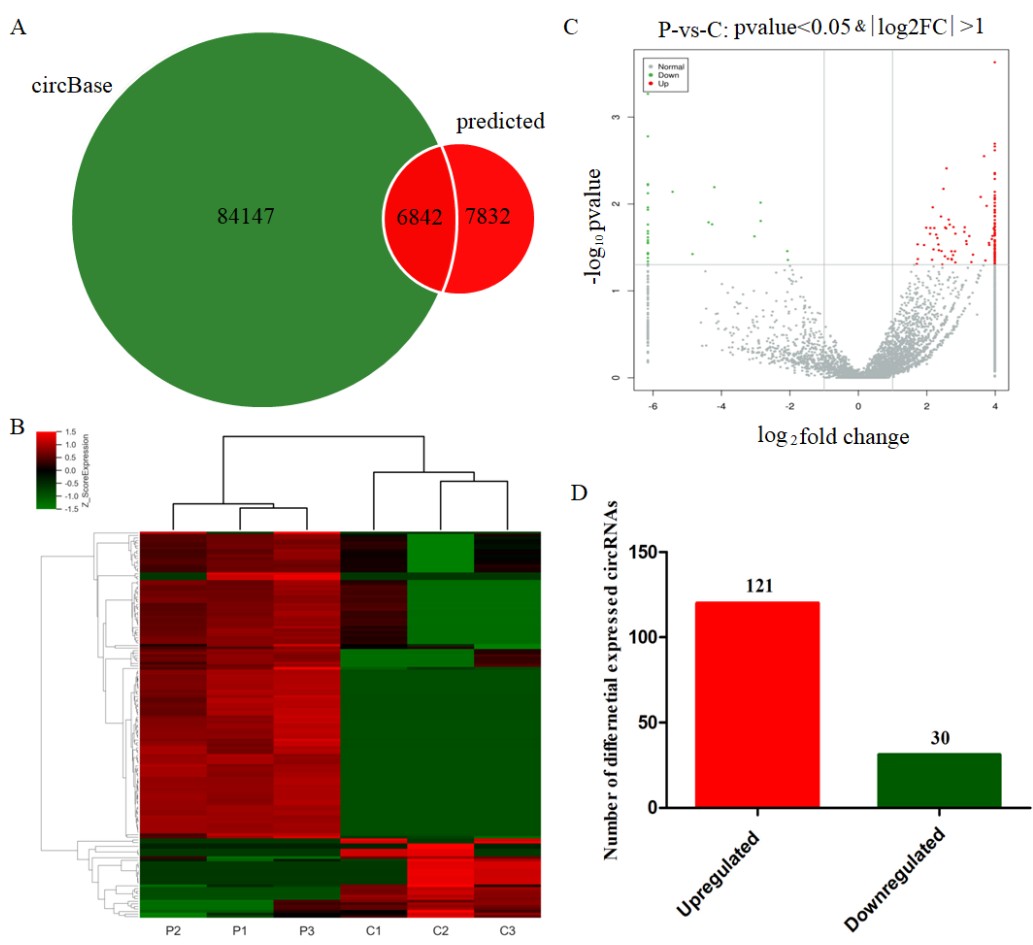

**Figure 1** **Identification of differentially expressed circRNAs from PE patients.** (A) Total circRNAs and known circRNAs compared with circBase. (B) Heatmap showing the differentially expressed circRNAs from the PE patients (P1, P2 and P3) compared with normal healthy individuals (C1, C2 and C3). (C) Volcano plot showing the differentially expressed circRNAs from the six samples. (D) Statistical results of differentially expressed circRNAs in PE patients compared with healthy individuals.

complex disassembly, negative regulation of toll-like receptor signaling pathway, negative regulation of protein complex disassembly, negative regulation of interleukin-6 production, response to exogenous dsRNA and negative regulation of innate immune response. The top 30 molecular function related GO terms that were found to be enriched in upregulated circRNAs included: protein heterodimerization activity, SH3 domain binding, receptor binding, cadherin binding, RNA binding, magnesium ion binding, protein tyrosine phosphatase activity, protein heterodimerization activity, ubiquitin protein ligase binding and protein kinase binding (Fig. 2A). The top 30 biological processes related GO terms that were found to be significantly enriched in downregulated circRNAs included: regulation of immune response, cell surface receptor signaling pathway, immune response, adaptive immune response, neutrophil degranulation and transcription. The top 30 molecular function related GO terms that were found to be significantly enriched in downregulated

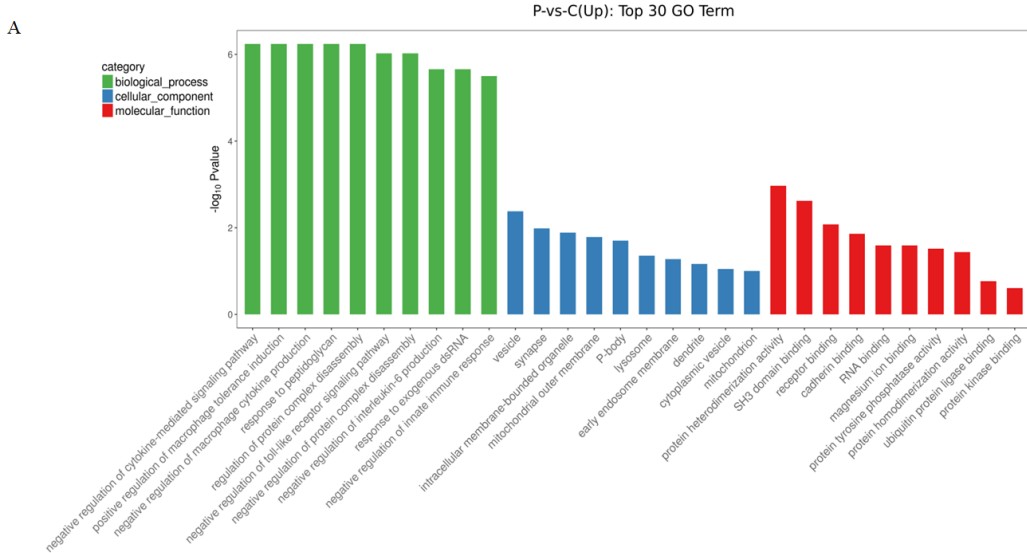

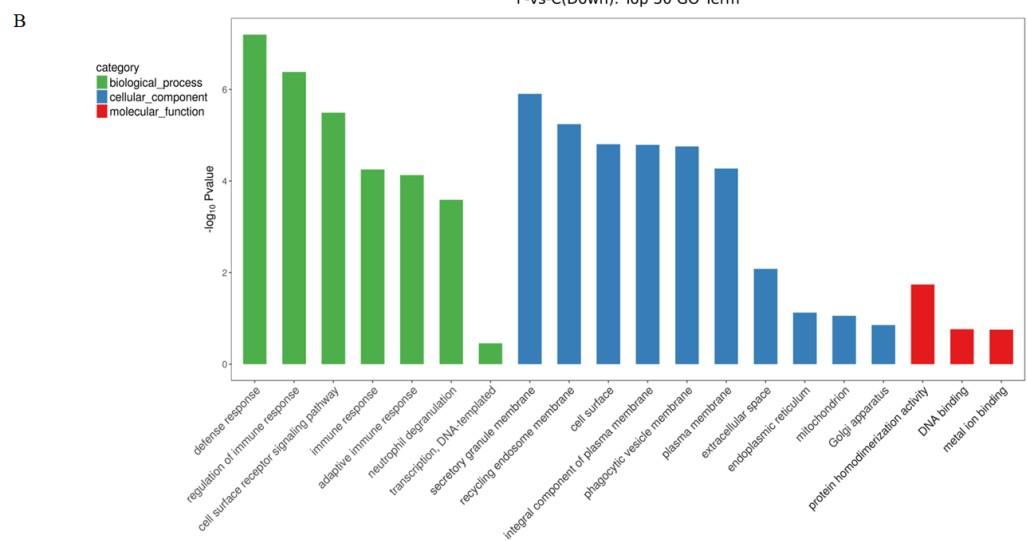

**Figure 2  GO enrichment analysis of differentially expressed circRNAs using their parental transcripts.** (A) GO enrichment analysis of upregulated circRNAs. (B) GO enrichment analysis of downregulated circRNAs.

circRNAs included: protein homodimerization activity, DNA binding and metal ion binding (Fig. 2B). From the GO enrichment analysis of the source transcripts of DE circRNAs, many important GO terms were revealed including immune response and protein homodimerization activity.

## KEGG enrichment of DE circRNAs

The source transcripts of DE circRNAs were further used to identify significantly enriched KEGG pathways in the list of DE circRNAs. 55 KEGG pathways (FDR < 0.05) were

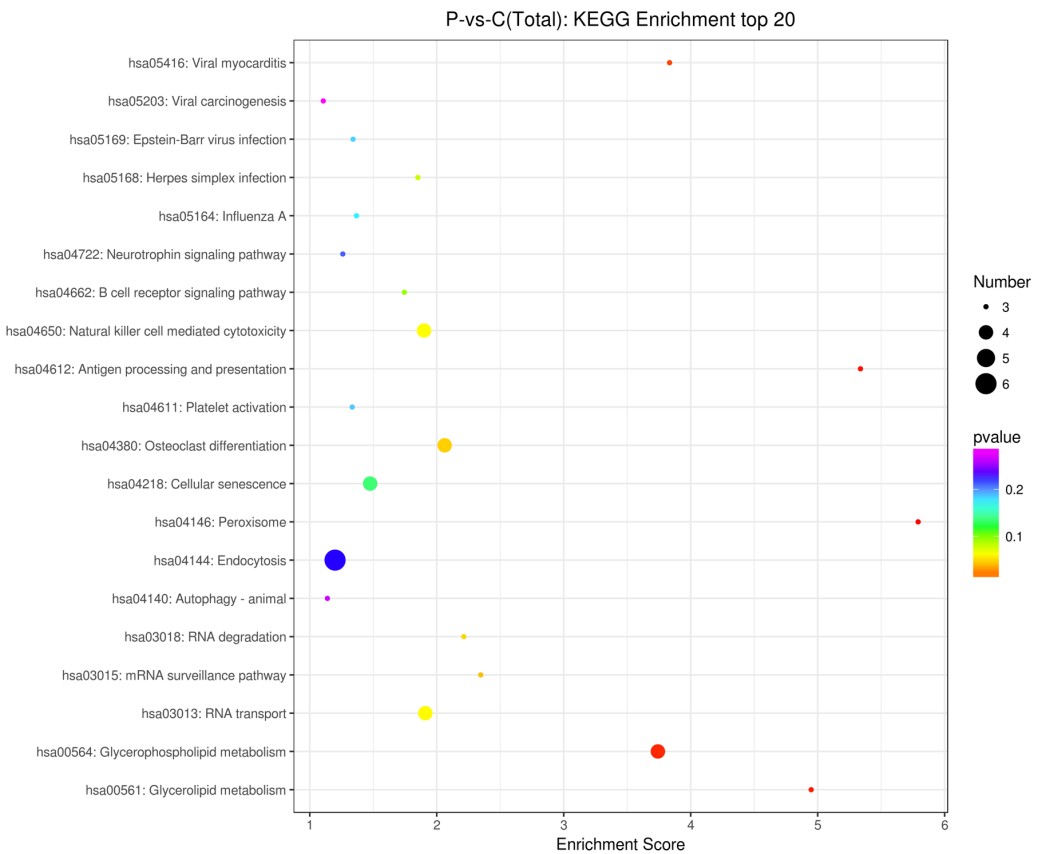

**Figure 3** KEGG enrichment analysis of differentially expressed circRNAs using their parental transcripts.

revealed to be significantly enriched in the DE circRNAs. The top 20 enriched pathways enriched in upregulated circRNAs included: glycerolipid metabolism, glycerophospholipid metabolism, mRNA surveillance pathway, RNA degradation, RNA transport, Influenza A, neurotrophin signaling pathway and endocytosis (Fig. 3). The top 20 enriched pathways in downregulated circRNAs included osteoclast differentiation and human cytomegalovirus infection (Fig. 3). These results revealed that some DE circRNAs were enriched in lipid metabolism and virus infection.

## Validation of circRNA expression

A set of DE circRNAs(hsa_circ_0001289, hsa_circ_0008311, hsa_circ_0004960, hsa_circ_0003753, hsa_circ_0001861, hsa_circ_0006719 and hsa_circ_0005806) was selected to validate the findings of the differential expression analysis using real-time PCR. Similar trends in the expression profile of all these circRNAs were identified between real-time PCR and high-throughput sequencing data (Figs. 4A–4G). Moreover, the junction sites of these circRNAs were further validated with reverse transcription PCR and Sanger sequencing (Figs. 4H–4N). These results confirmed the validity of the sequencing data from the PE patients.

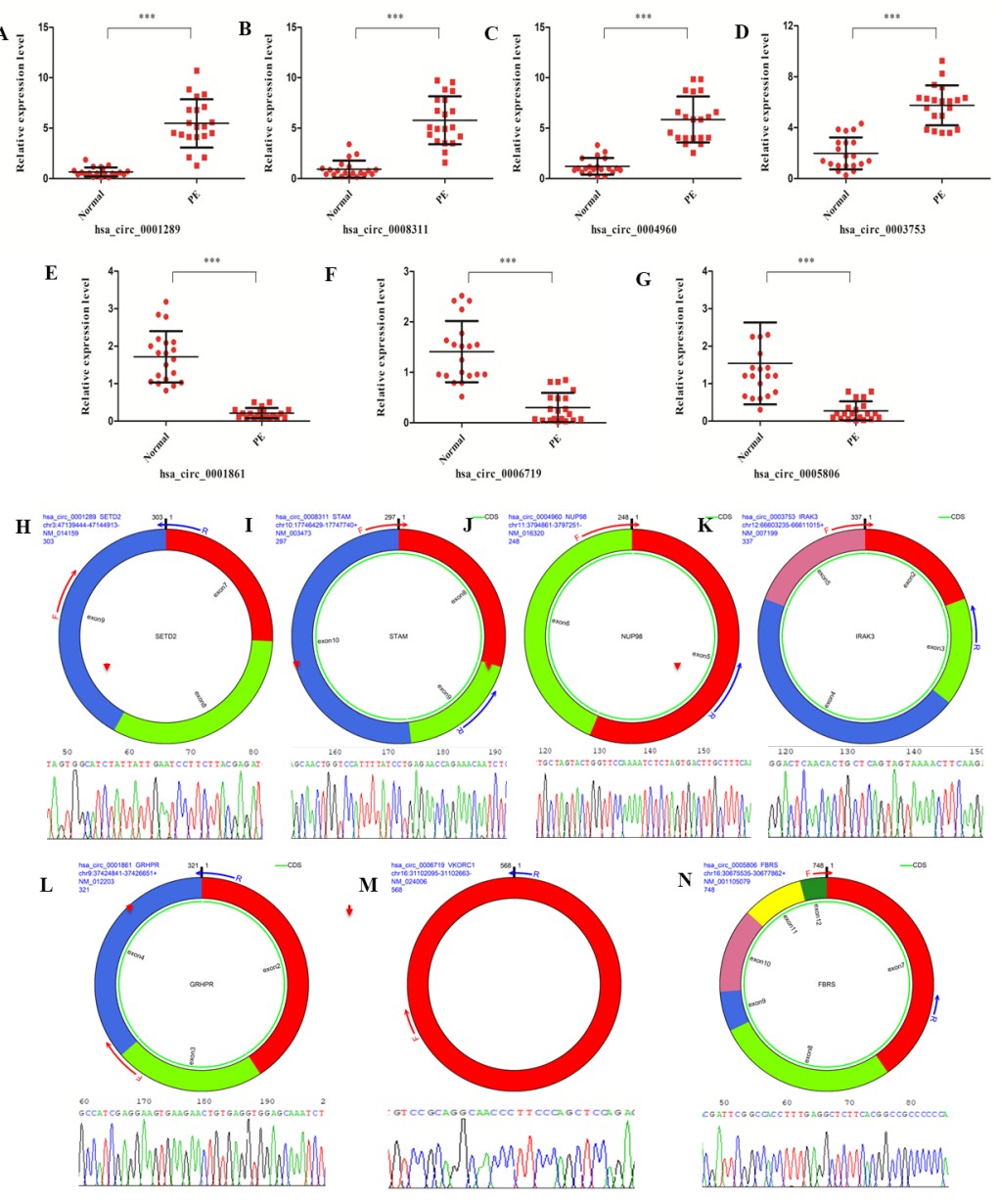

**Figure 4  circRNAs validation with real-time PCR and reverse transcription PCR.** (A–G) Validation of differentially expressed circRNAs (hsa_circ_0001289, hsa_circ_0008311, hsa_circ_0004960, hsa_circ_0003753, hsa_circ_0001861, hsa_circ_0006719 and hsa_circ_0005806) from the 20 PE patients and healthy individuals with real-time PCR. (H–N) Validation of junction site of the circRNAs with reverse transcription PCR.

## CircRNA-miRNA interaction analysis

The expression level of mRNAs is regulated by circRNAs through miRNA mediation. Therefore, a comprehensively analysis of the interactions between circRNA and miRNA will help to explore the potential regulatory mechanisms of circRNAs in PE. A circRNA-miRNA network was constructed based on the differentially expressed circRNAs and differentially
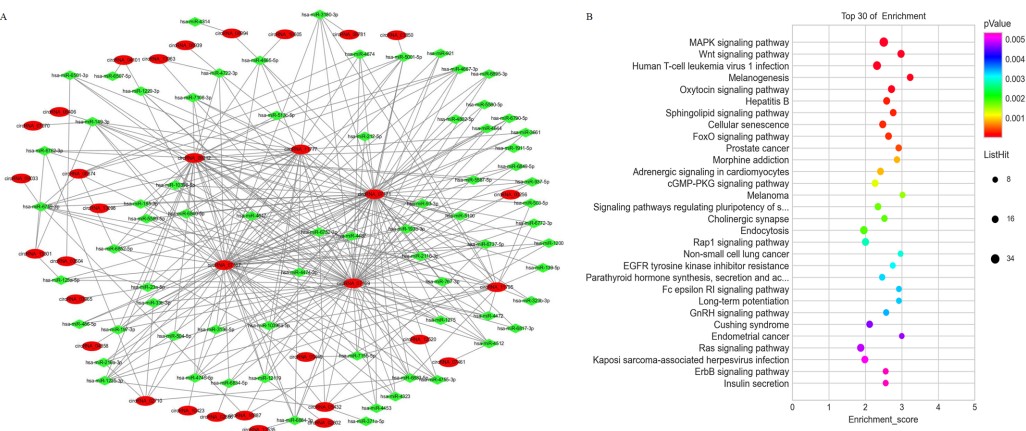

**Figure 5 The co-expression network of circRNAs and miRNAs and the associated signaling pathways analysis with the targets of co-expressed miRNAs.** (A) The network of differentially expressed circR-NAs with miRNAs. (B) The enriched signaling pathways were predicted with the using the targets of co-expressed miRNAs.

expressed miRNAs identified from the PE and visualized with Cytoscape (Fig. 5A). The results showed that each circRNA could sponge several miRNAs and each miRNA can have several target circRNAs. KEGG pathway enrichment analysis was conducted on the mRNA which are predicted as miRNA targets to provide further insights into the biological processes of mRNAs mediated by the differential expressed circRNA through miRNAs. The results indicated that the miRNA target transcripts were enriched in signaling pathway of calcium, insulin secretion, prolactin, biotin metabolism and GnRH (Fig. 5B).

## circRNA-miRNA-mRNA network re-construction

Two ceRNAs networks were re-constructed according to the targeting/interaction relationship between circRNAs/miRNAs and miRNAs/mRNAs to investigate the potential regulatory mechanism of DE circRNAs in the etiology and pathogenesis of PE. The *p*-values threshold for the of correlation coefficient of the relative expressions was set to 0.05, and the threshold for absolute value of Pearson correlation coefficient was set to 0.7. One subnetwork included one upregulated circRNA, four downregulated miRNAs and five upregulated mRNAs (Fig. 6A), and another subnetwork included 10 downregulated circRNAs, 21 upregulated miRNAs and 15 downregulated mRNAs (Figs. 6B, 6C). These interacted circRNA-miRNA-mRNA networks could contribute to further exploring the regulatory mechanism of DE circRNAs in the pathogenesis of PE.

## DISCUSSION

Recent studies have accumulated evidence that circRNAs, a novel class of non-coding RNAs, are associated with several crucial biological roles, such as miRNA sponges, RNA-binding protein and translation (*Kristensen et al., 2019*). Among roles, the one of miRNA sponges was comprehensively identified in the progression and the pathogenesis of diverse of cancers. In these cases circRNAs were found to act as ceRNAs via adsorbing miRNAs to

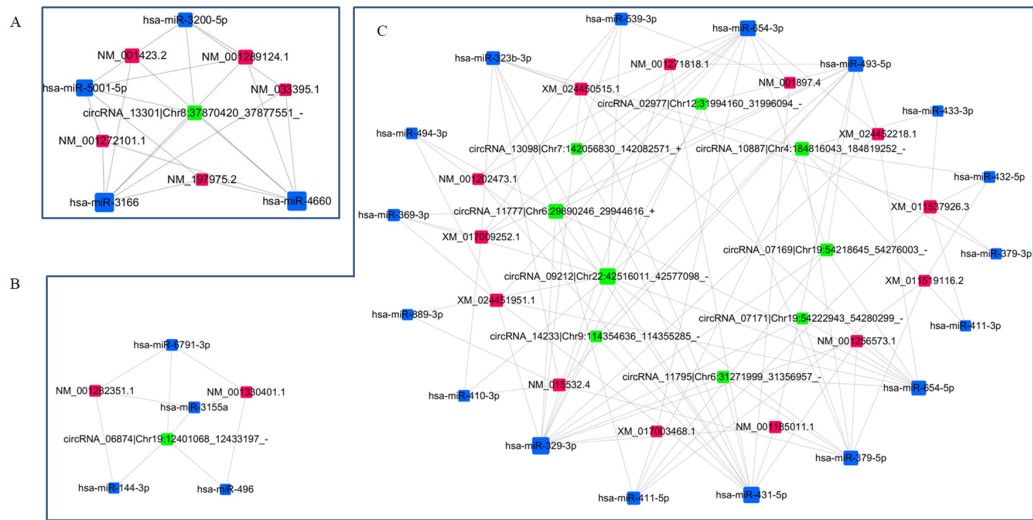

**Figure 6  Re-construction of ceRNA networks with differentially expressed circRNAs, miRNAs and mRNAs.** (A) The ceRNA network including upregulated circRNAs, downregulated miRNAs and upregulated mRNAs. (B, C) The ceRNA network including downregulated circRNAs, upregulated miRNAs and downregulated mRNAs.

regulate the expression levels of their target genes. In addition, numerous circRNAs have been suggested as novel biomarkers for early diagnosis of diseases (*Tian et al., 2019*; *Wu et al., 2019*). However, to the best of our knowledge, there exist limited reports about the role of circRNAs in the etiology and pathogenesis of PE. In present study, the DE patterns of circRNAs were explored in blood samples of PE patients compared to normal pregnant women with 121 circRNAs found to be upregulated and 30 downregulated in PE patients. These statistically significant differentially expressed circRNAs in PE patients indicated that circRNAs might play an important regulatory role in the pathogenesis of PE. Furthermore, using the parental transcripts of these circRNAs to explore the potential molecular functions and signaling pathways, we identified that the DE circRNAs were significantly enriched to various biological processes and signaling pathways.

PE is a serious and common pregnancy complication, which is considered one of the causes for increased morbidity and mortality in pregnancies (*Antwi et al., 2013*). It is known that many biological processes (immune maladaptation, inadequate placental development, trophoblast invasion, placental ischemia, oxidative stress and thrombosis) have been associated with the etiology and the pathogenesis of PE (*Zhou et al., 2018*), but the mechanism underlying PE development was still unclear. circRNAs is a novel type short non-coding RNAs with circular structure and they have been reported to play crucial roles in the progression and pathogenesis of diverse diseases (*Zhang et al., 2017*). The ceRNA hypothesis was comprehensively applied to explore the molecular pathogenesis of various diseases (*Salmena et al., 2011*) including PE. 180 DE circRNAs were identified from the placentae of severe preeclampsia, and many of them were associated to the vasodilation and the regulation of blood vessel size (*Deng et al., 2019*). 49 DE circRNAs were identified

in the placental tissues of PE, and this group of DE circRNAs was enriched in in MAPK signaling pathway (*Ou et al., 2019*). Another 49 DE circRNAs were identified from the placental tissue of PE women compared with healthy pregnant women with these playing important roles in cellular regulation via sponging miRNAs (*Zhou et al., 2018*). Circulating circRNAs may also have some predictive/diagnostic value for PE (*Zhang et al., 2016*). Most of these studies were carried out in the placental tissue of PE and the expression levels of some circRNAs found to be significantly changing in the development of PE patients. Much fewer studies involve analysis of peripheral blood samples from PE patients.

In the present study, we constructed the ceRNAs to reveal the potential molecular roles of dysregulated circRNAs in the PE patients using RNA sequencing data from the PE patients vs. healthy pregnant women. circRNA_13301 was the only one upregulated circRNA in ceRNA network, which contained five upregulated mRNAs (centrosomal protein 295, epithelial membrane protein 1, butyrophilin like 3, matrix metallopeptidase 19, LIM and calponin homology domains 1) being targeted by four miRNAs (hsa-miR-3200-5p, hsa-miR-4660, hsa-miR-5001-5p and hsa-miR-3166). Another ceRNA network contained 10 downregulated circRNAs, 21 upregulated miRNAs and 15 downregulated mRNAs. KEGG pathway enrichment analysis found that these downregulated mRNAs were enriched into nicotine addiction, cholinergic synapse, cell adhesion molecules and calcium signaling pathway. These novel mRNAs mediated by differential expressed circRNAs by sponging miRNAs, which roles played in the pathogenesis will be validated in the further research.

Our study is the systematic profiling of ceRNAs in leukocyte of PE patients and revealed the global ceRNA network integration in PE. These data will provide novel clues to understand the etiology and pathogenesis of PE from the dysregulated circRNAs.

## CONCLUSIONS

In summary, our findings revealed the the circRNA expression patterns in PE patients and their possible regulatory mechanisms. This study provides useful information for exploring the potential roles of circRNAs in the etiology and pathogenesis of PE.

## ACKNOWLEDGEMENTS

We thank Lin He's Academician Workstation of New Medicine and Clinical Translation for its guidance and help. The authors would like to thank Dr. Qidong Zu (OE Biotech, Inc., Shanghai, China, http://www.oebiotech.com/) for assistance with the bioinformatics analysis of whole-transcriptome sequencing.

### Funding

The project was funded by support from Medical Health Science and Technology Project of Zhejiang Provincial Health Commission (2020KY962), Jiaxing Municipal Public Welfare Research Plan (2021AD30129, 2021AD30131), The Natural Science Foundation of the Jiangsu Higher Education Institutions of China (19KJB320005). The Promoting Science

and Education of Suzhou (KJXW2017026 and SZYJTD201709). The funders had no role in study design, data collection and analysis, decision to publish, or preparation of the manuscript.

## Grant Disclosures

The following grant information was disclosed by the authors:
Medical Health Science and Technology Project of Zhejiang Provincial Health Commission: 2020KY962.
Jiaxing Municipal Public Welfare Research Plan: 2021AD30129, 2021AD30131.
The Natural Science Foundation of the Jiangsu Higher Education Institutions of China: 19KJB320005.
The Promoting Science and Education of Suzhou: KJXW2017026, SZYJTD201709.

## Competing Interests

The authors declare there are no competing interests.

## Author Contributions

- Zepeng Ping conceived and designed the experiments, analyzed the data, authored or reviewed drafts of the paper, and approved the final draft.
- Ling Ai conceived and designed the experiments, prepared figures and/or tables, and approved the final draft.
- Huaxiang Shen and Ye Song analyzed the data, prepared figures and/or tables, and approved the final draft.
- Xing Zhang and Huling Jiang conceived and designed the experiments, performed the experiments, authored or reviewed drafts of the paper, and approved the final draft.

## Human Ethics

The following information was supplied relating to ethical approvals (i.e., approving body and any reference numbers):

Maternity and Child Health Care Affiliated Hospital, Jiaxing University granted Ethical approval to carry out the study within its facilities (Ethical Application Ref: 2019-47).

## DNA Deposition

The following information was supplied regarding the deposition of DNA sequences:
Sequences are available at the NCBI Sequence Read Archive: PRJNA665923.

## Data Availability

The raw numeric data for real-time PCR are available in the Supplemental File.

## Supplemental Information

Supplemental information for this article can be found online at http://dx.doi.org/10.7717/peerj.11299#supplemental-information.

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
