# Peer review of "Identification and comparison of circular RNAs in preeclampsia"

_PeerJ, doi:10.7717/peerj.11299_

## Round 0.1 · original submission · Major Revisions

We have received critical comments from three reviewers. Please describe the functional role of circRNAs in more detail, present novel findings.

Despite the critical remarks, as the academic editor, I encourage the authors to revise and resubmit this work.

Reviewer 1 ·

Basic reporting

This manuscript is clear and unambiguous and the literature reference is properly cited. The figures and tables are professionally generated.

Experimental design

This manuscript fits the Aims and Scope of the journal. The research question is clearly defined. Basically it stated how research fills the reported knowledge gap. The described Methods contained details to replicate.

Validity of the findings

The impact and novelty of this manuscript is limited. The data are statistically sound and the conclusions are stated.

Additional comments

In this study Jiang and colleagues investigated the pathogenesis the pregnancy-specific syndrome Preeclampsia (PE) through RNA sequencing. The authors carried out RNA-seq analysis of rRNA-depleted RNA samples derived from 20 patients and 20 normal wowen. Compared with the normal, the authors identified 151 differentially expressed circRNAs with 121 upregulated and 24 downregulated ones. Gene ontology analysis were performed and came out with several enriched biological processes and pathways in PE patients. Finally the authors constructed two ceRNA networks and found some either upregulated or downregulated miRNAs. The authors concluded their findings revealed the potential regulatory mechanisms of PE. Overall, the authors have done quite a lot of work, including both experimental and computational work. However, several issues are pending to be addressed.

1. The results do not convincingly support the conclusion. That is, the authors claimed they revealed mechanisms of PE. How circRNAs/miRNAs identified in this study could facilitate the understanding of PE pathogenesis? It was not clearly mentioned in Results, nor in Discussion. The authors should provide more information regarding it.
2. Line 87: the authors only described total RNA extraction, but how rRNA-depleted RNA was prepared was actually not mentioned, particularly this part was subtitled with rRNA-depleted RNA sequencing.
3. Line106: the criteria for identifying differentially expressed circRNAs were p value threshold of 0.05 and fold change threshold equal to 2, but in line 139 they were said as adjusted p value < 0.05; fold change, > 2 or < 140 0.5. This inconsistency needed to be clarified.
4. Line 110: ceRNA (competing endogenous RNA),the full name of such abbreviation should be provided when it was firstly mentioned in the main text.

Reviewer 2 ·

Basic reporting

The authors identified differentially expressed circRNAs. A functional analysis of expressed circRNAs in patients with PE was performed using the Gene Ontology and KEGG databases. The authors determined the circRNA - microRNA (miRNA) interactions. Two networks were built, one containing positively and the other negatively regulated miRNAs.

No sources for 2020. Only references to the participation of circRNAs are provided without specifying the results of the cited literature. That is, the specific purpose of the study is not substantiated.

Experimental design

Results of using programs Miranda and Diana not provided. That is, there is no evidence of miRNA/circRNAs and mRNA interaction. The authors need to provide schemes for the interaction of miRNA with circRNAs and miRNA with mRNA with the application of quantitative characteristics.

The degree of suppression of expression and upregulation of circRNAs in the two groups of patients is radically different (Fig 4), which raises doubts.

The quality of the figures is poor.

Validity of the findings

There are no novelty in the manuscript. In general, the manuscript contains a set of little related results and their discussion. As a result, the conclusion does not contain the key regulatory mechanisms out of 268 (269) possible regulatory mechanisms.

Additional comments

The manuscript is not recommended for publication.

Reviewer 3 ·

Basic reporting

In the present manuscript, the authors identify 14,674 circRNAs in normal pregnant women and PE patients. They found the 151 differential expressed circRNAs, performed functional and pathway enrichment analysis of these circRNAs, predicted potential interactions, and constructed ceRNA networks. Additionally, they validated 7 circRNAs and their expression.

The manuscript is good with minor mistakes. Numbers should use scientific notation, repeated periods in line 199 and 252, repeated space in line 216, words with underscore 249 and 254, and so on.

The quality of the figures are poor, it is hard to get useful information from it, especially figure 2 and figure 4.

Experimental design

In the introduction, the author briefly introduced the circRNA roles in diseases and circRNA progress in PE several diseases. However, this part needs more detailed description. Besides, it is better to mention the progress of ncRNA in PE.

In the manuscript, the authors provided some details of their methods. I don’t understand why the authors use DESeq tool (usually use counts) to perform differential expression analysis, especially the expression of circRNAs is calculated by RPM algorithm. The RPM algorithm is a normalized method which ignores transcript length, while the authors seem to have different ideas (line 98 to 99). Besides, please provide the version of CIRI.

The authors mentioned they constructed the circRNA-miRNA network based on the differentially expressed circRNAs and differentially expressed miRNAs. However, it seems no small RNA-seq in this project, how to get differentially expressed miRNAs?

Validity of the findings

no comment

Additional comments

In the Results, the authors should give the characterization of 14,674 identified circRNAs. Only comparing those circRNAs with circBase (not CircBase) can not visually display the characteristics of those 14,674 circRNAs, such as types, length, and distribution.

---

## Round 0.2 · Major Revisions

We have received contradictory reviews. The manuscript was revised, but the current version still needs major revision.

Please tone down the conclusion or bring new data to prove your conclusion.See remark #1 from Reviewer#1:
* * *
1. The results do not convincingly support the conclusion. That is, the authors claimed they revealed mechanisms of PE. How circRNAs/miRNAs identified in this study could facilitate the understanding of PE pathogenesis? It was not clearly mentioned in Results, nor in Discussion. The authors should provide more information regarding it.

Reviewer 1 ·

Basic reporting

This manuscript is clear and unambiguous.

Experimental design

The detailed information of rRNA depletion is still not provided.

Validity of the findings

Their findings are not very impressive. In particular, the mechanisms of PE is not well characterized.

Additional comments

The authors has partially addressed my concerns. However, the major concern in comment 1 is not clarified.

Reviewer 2 ·

Basic reporting

no comment

Experimental design

no comment

Validity of the findings

no comment

Additional comments

All the comments I made to the authors were taken into account.

Reviewer 3 ·

Basic reporting

The authors answered my prevous questions. But less improvement. the novelty is not high. The quality of the figures are still not good. and there are a number of lingustic mistakes. the references are not uniformed.

Experimental design

the authors listed a publications on ncRNA in PE. But less discussions of them comparing with this work. Again, the circRNA-miRNA network was based on the differentially expressed circRNAs and differentially expressed miRNAs. But without small RNA-seq, even not published, how can we believe the scitific correntness of your result?

Validity of the findings

NN

Additional comments

NN

---

## Round 0.3 · Minor Revisions

The manuscript got positive estimates from the reviewers. However there are some minor comments regarding English presentation and figures quality.

I suggest you check Abstract again and update the concluding phrase (lines 34-35)
“These results will provide useful information...” is too common phrase. Put instead from main text (from lines 271-275):
We constructed the ceRNAs (competing endogenous RNA) to reveal the potential molecular roles of dysregulated circRNAs in the PE patients using RNA sequencing data from Jiaxing University. circRNA_13301 was the only one upregulated circRNA in ceRNA being targeted by four miRNAs.
(just suggestion - make the resulting statement in the Abstract)

Then, avoid extra abbreviation “DE”
differentially expressed (DE) - in the Abstract since it used only once.
Give the abbreviation ceRNAs in full as competing endogenous RNA at the point of first mention in the text.

In Table 1 move the line” Gestational weeks” to the second row, after the “Age (years)” line to show that common parameters in PE and healthy patients not differ (>0.05)
And the rest lines (<0.01) are together (may underline or mark by bold font too)

Table 2 “Primers for validation of circRNAs” is not informative - try move it to supplement or update by some information (like length in bp). At least format by standard font size.

Figure 1: change in the panel A “circbase” to “circBase” and “predict” to “predicted”
In panel B: make fonts larger on the axes (it is too small now)
Change colors - match colors to panel D - both green and red or blue and red.
I think panel D (just 2 numbers in the histogram) is not informative at all. May remove panel D.
Figure 2 could be updated. The colors are not easy to see. May make GO categories names in axis X in standard black fonts, not in green/blue.
It is better to show less top GO categories, not 30, it will be easier to read and understand.
Panel B in Figure 2 has less than 30 categories. Then update the title.

Figure 3 - font size in axes X and Y is too small. Panel B is not good - it has only two dots. MAy remove it, or try to join with panel A (put these dots for the downregulated over, in other shape (not circles), or mark by some other way)
Figure 4 -
It is kind of redundancy. But may keep it, just make fonts larger. For panels L,M,N - make it in same size as for H-K.
Figure 5
Panel A - too small font, mRNA names are not readable
Panel B (histogram) -
Please remove shadow/volume effects/ Increase font size.
Some categories (pathways) are not significant (>0.01). You may remove it to make more compact histogram.
Figure 5.
Panel B has too small network (and small fonts). Match it in size with panel A.
It is my suggestion.
Need to improve readability of the figures anyway.
Next round we may not ask reviewers to check to process it faster. Waiting your revised manuscript.

Reviewer 1 ·

Basic reporting

No comment.

Experimental design

No comment.

Validity of the findings

No comment.

Additional comments

The revisions look OK.

Reviewer 2 ·

Basic reporting

no comment

Experimental design

no comment

Validity of the findings

no comment

Additional comments

The conclusions are appropriately stated, connected to the original question investigated, and those supported by the results.

Reviewer 3 ·

Basic reporting

the revision is fine. the quality of the figures are not good.

Experimental design

nn

Validity of the findings

nn

Additional comments

although the findings are not so significant and research originality is not high, the revision is acceptable for potential publication in PeerJ. the english and figures requires more improvement.

---

## Round 0.4 · accepted · Accept

Thanks again for the text update and detailed answer. I have no more remarks; I endorse the manuscript publication.